# Effect of Yeast Fermentation on the Physicochemical Properties and Bioactivities of Polysaccharides of *Dendrobium officinale*

**DOI:** 10.3390/foods12010150

**Published:** 2022-12-28

**Authors:** Hang Chen, Xueqin Shi, Lanyan Cen, Lin Zhang, Yifeng Dai, Shuyi Qiu, Xiangyong Zeng, Chaoyang Wei

**Affiliations:** 1Key Laboratory of Fermentation Engineering and Biological Pharmacy of Guizhou Province, School of Liquor and Food Engineering, Guizhou University, Guiyang 550025, China; 2Sichuan Langjiu Co., Ltd., Luzhou 645423, China; 3Key Laboratory of Plant Resource Conservation and Germplasm Innovation in Mountainous Region (Ministry of Education), Institute of Agro-bioengineering, College of Life Sciences, Guizhou University, Guiyang 550025, China

**Keywords:** *Dendrobium officinale*, polysaccharides, fermentation, antioxidant activity, immune activity

## Abstract

Fermentation is an effective method for enhancing the biological activity of polysaccharides, but research on its effect on *Dendrobium officinal* polysaccharides is rare. In this study, the effects of mono-fermentation (*Saccharomyces cerevisiae* FBKL2.8022, *Sc*; *Wickerhamomyces anomalous* FBKL2.8023, *Wa*) and co-fermentation (*Sc+Wa*) on the physicochemical properties and bioactivity of *Dendrobium officinal* polysaccharides were investigated. Meanwhile, the polysaccharide (DOP) obtained from *Dendrobium officinale* was used as a control. Four homogeneous polysaccharides were obtained by isolation and purification and named DOSCP, DOWAP, DOSWP, and DOP. The results showed that DOSCP, DOWAP, DOSWP, and DOP consisted of mannose and glucose with ratios of 3.31:1, 5.56:1, 2.40:1, and 3.29:1, respectively. The molecular weights (Mws) of the four polysaccharides were 25.73 kDa, 15.01 kDa, 17.67 kDa, and 1268.21 kDa. The antioxidant activity of DOSCP, DOWAP, and DOSWP was better than that of DOP. Additionally, all four polysaccharides were able to reduce the inflammatory response of LPS-induced RAW 264.7 macrophages in the mice without a significant difference. Yeast fermentation significantly reduced the molecular weight and improved the antioxidant activity of *Dendrobium officinale* polysaccharides, indicating a potential way to improve its antioxidant activity.

## 1. Introduction

*Dendrobium officinale* belongs to the orchid family, and is a traditional herb used for the treatment or prevention of many diseases. Pharmacological studies have shown that *Dendrobium officinale* has hepatoprotective, antitumor, immunomodulatory, hypoglycemic, gastrointestinal protective, anti-inflammatory, and vasodilatory effects [1]. These biological activities are attributed to abundant bioactive components, including polysaccharides, phenanthrenes, flavonoids, alkaloids, biphenyl groups, and other compounds. Recent studies have demonstrated that *Dendrobium officinale* polysaccharides possess antitumor, hepatoprotective, and antiglycation properties [2]. Notably, many bioactivity tests are mainly based on crude polysaccharides, which makes it difficult to prove the exact active substance, and the study of polysaccharide structure–activity relationships has yet to be further investigated. Therefore, studying the biological activity of purified polysaccharides and their mechanism of action is valuable. Studies have shown that the monosaccharide composition of *Dendrobium officinale* polysaccharides is mainly mannose, glucose, galactose, xylose, arabinose, and rhamnose [2]. The *Dendrobium officinale* polysaccharide DOP has D-mannose, D-glucose, and D-galactose with molar ratios of 3:2:1 and a relative molecular weight of 1.89 × 10^5^ Da, according to Sun et al. [3]. Huang et al. [4] reported that DOPA-1 and DOPA-2 (*Dendrobium officinale* stem polysaccharides) consisted of 1,4-β-D-Man*p*, 1,4-β-D-Glc*p*, and O-acetyl. It can stimulate splenocytes, T-lymphocytes, and B-lymphocytes, promote cell viability and NO production in RAW264.7 macrophages, and protect RAW264.7 macrophages from H_2_O_2_-induced oxidative damage.

The application of high molecular weight polysaccharides is limited due to their apparent viscosity, water solubility, structure, and conformation. In contrast, low molecular weight polysaccharides have higher activity and are more significant for human health [5]. Therefore, chemical methods [6], enzymatic methods [7], and microbial methods [8] are widely applied for polysaccharide extraction and degradation. In recent years, the extraction of plant polysaccharides by microbial fermentation has become a popular method and has received much attention [9]. Fermentation is one of the best biomodification methods for enhancing the biological activity of polysaccharides. The biological activity of the active ingredients in plants can be improved or biotransformed by the action of microorganisms on the raw material during fermentation. For instance, Ai et al. [9] used *Saccharomyces cerevisiae* GIW-1 to ferment ginseng, which significantly increased the yield of ginseng polysaccharides by 4.10%, as well as its antioxidant and anti-inflammatory activity. Chen et al. [10] fermented wheat bran with *Saccharomyces cerevisiae* and *Bacillus subtilis*, and the fermentation process increased the content of total polysaccharides. After fermentation, the polysaccharide structure and monosaccharide composition changed, the molecular weight decreased, and free radical scavenging capacity and reduction ability significantly improved. Li et al. [11] used lactic acid bacteria to ferment *Dendrobium officinale* polysaccharides. They found that fermentation significantly altered the content of total sugars and reducing sugars, the molar ratio of monosaccharides, and the distribution of polysaccharide molecular weights.

Fermentation is an effective way to reduce the viscosity by breaking down polysaccharides, which may be due to the intramolecular interactions or random winding of molecular chains reduced after fermentation [12]. Reducing the molecular weight of polysaccharides by fermentation improve their biological activity. However, there are few studies on the impacts of fermentation on the structure and bioactivity of *Dendrobium officinale* polysaccharides. Therefore, it is necessary to study the changes in the structure and biological activity of *Dendrobium officinale* polysaccharides during and after fermentation. The two yeast strains used in this study, *Saccharomyces cerevisiae* FBKL2.8022 and *Wickerhamomyces anomalous* FBKL2.8023, were screened and separated from the traditional block koji in Guizhou and stored in our lab (Guizhou Provincial Key Laboratory of Fermentation Engineering and Biopharmaceuticals). They have a low yield of higher alcohols and can be used to ferment rice wine. They not only increase the biological activity of *Dendrobium officinale*, but also reduce the yield of higher alcohols in the liquor [13]. In this study, we extracted polysaccharides from fermented *Dendrobium officinale* rice wine and compared the changes in the antioxidant properties of *Dendrobium officinale* polysaccharides before and after fermentation and the changes in inhibitory effects on inflammatory cytokines (TNF-α, NO, IL-6, IL-1β, and IL-10) produced by lipopolysaccharide-treated RAW264.7 cells based on the physical and chemical properties of *Dendrobium officinale* polysaccharides before and after fermentation. The results of this study will provide a scientific basis for the in-depth study and application of structural changes and antioxidant and anti-inflammatory activities of *Dendrobium officinale* polysaccharides before and after fermentation.

## 2. Materials and Methods

### 2.1. Materials and Reagents

*Dendrobium officinale* was purchased from Longanba Green Spring Agricultural Development Co., Ltd. (Guiding, China). The samples were dried at 60 °C, crushed, passed through an 80-mesh sieve and set aside.

*Saccharomyces cerevisiae* FBKL2.8022 *(Sc*) and *Wickerhamomyces anomalous* FBKL2.8023 (*Wa*) were derived from the Key Laboratory of Fermentation Engineering and Biological Pharmacy of Guizhou Province. The CCTCC NO. was M2019406 and M2019412, respectively [13]. Monosaccharide standards were purchased from Aladdin Reagent Co., Ltd. (Shanghai, China), including mannose, ribose, rhamnose, glucuronic acid, galacturonic acid, glucose, galactose, xylose, arabinose, and fucose. Amyloglucosidase and anhydrous D-glucose were purchased from Beijing Solarbio Science & Technology Co., Ltd. (Beijing, China). NO kits were purchased from Beyotime Biotechnology Co., Ltd. (Shanghai, China). TNF-α, IL-1β, IL-6, and IL-10 ELISA kits were purchased from Shanghai Jianglai Biological Co., Ltd. (Shanghai, China). The relative Mw of polysaccharides was calculated according to calibration curves of the dextran standards purchased from Sigma-Aldrich (St. Louis, MO, USA). DEAE-Sepharose Fast Flow (DEAE-QFF) and Sephacryl S-300 HR were purchased from GE Healthcare. A G4000SWXL column (7.8 × 300 mm, Tosoh Co., Ltd., Tokyo, Japan) was used. The other chemicals and reagents were analytical grade.

### 2.2. Preparation and Purification of Polysaccharides

The method of Wang et al. was used for the rice wine fermentation process [13]. A 250 mL triangular flask was used to hold50 g of glutinous rice after it had been soaked in 250 mL of distilled water overnight, drained, and sterilized at 121 °C for 40 min. Then, it was mixed with 75 mL of sterile water, 0.56% amyloglucosidase, and 1.89% D-glucose anhydrous and placed in a water bath at 60 °C for 30 min. Yeast seed solution that was inoculated at a concentration of 1 × 10^6^ cells/mL and 5% *Dendrobium officinale* powder were added into the broth simultaneously. Fermentation was conducted at a constant temperature of 30 °C, and the broth was weighed and recorded every 24 h. When the weight loss of CO_2_ was less than 0.2 g/d, the fermentation was finished. The rice wines were named DOSC, DOWA, and DOSW according to their inoculated yeast strains, where DOSW was mixed yeast fermentation with a ratio of 1:10 (*Sc*: *Wa*).

The extraction methods of the four polysaccharides are listed in Table 1. The extraction of *Dendrobium officinale* polysaccharides was performed using the hot water extraction method; that is, *Dendrobium officinale* powder was extracted at a material-liquid ratio of 1:40 g/mL, a temperature of 90 °C and a time of 2 h. After cooling, the extract was centrifuged for 10 min at 8000 rpm. The broth of *Dendrobium officinale* rice wine was centrifuged (10 min at 8000 rpm), and the supernatant was taken to prepare rice wine polysaccharides. Fermentation residues were used to extract crude polysaccharides according to the above extraction method. Anhydrous ethanol with a fourfold volume was mixed with the supernatant overnight. Then, the precipitate was centrifuged (8000 rpm, 5 min), collected, re-dissolved, and lyophilized to obtain crude polysaccharide. The deproteinization, isolation, and purification processes of crude polysaccharides were performed according to our previous method [14]. The polysaccharide content was determined by the phenol-sulfate method [15]. The protein content was determined by the Bradford method [16].

### 2.3. Determination of Monosaccharide Composition and Molecular Weight (Mw)

The precolumn derivatization approach was used to determine the monosaccharide composition of polysaccharides [17]. The sample was dissolved by adding 3 mg of polysaccharide to 1 mL of distilled water and hydrolyzed with 1 mL (4 mol/L) of trifluoroacetic acid (TFA) at 110 °C for 6 h. The residual TFA was then removed by adding methanol. 1 mL of hydrolysis product was added to distilled water for derivatization. After derivatization, the derivatives filtered through a 0.22-μm membrane were detected by an Agilent Eclipse XBD-C18.

The relative Mw of polysaccharides was measured using a high-performance liquid chromatography system equipped with a Waters-2414 differential refractive index detector (Waters Co., Milford, MA, USA). A G4000SWXL column (7.8 × 300 mm, Tosoh Co., Ltd. Tokyo, Japan) was used. The column temperature was 40 °C, and 0.1 M sodium nitrate solution was used as the eluent. The relative Mw of polysaccharides was calculated according to calibration curves (Log Mol Wt = 1.27e − 8.10e^−1^ T, R^2^ = 0.9943) of the dextran standards (Sigma Co., St Louis, MI, USA), including 667,000 Da, 413,000 Da, 76,900 Da, 43,500 Da, 10,500 Da, and 5000 Da.

### 2.4. Antioxidant Activities Assay

#### 2.4.1. DPPH Scavenging Activity

We referred to the method of Chen et al. [18] and made modifications. Polysaccharides were prepared into 0.2, 0.4, 0.6, 0.8, 1.0, and 2.0 mg/mL solutions. Hence, 2 mL of polysaccharides solution and 2 mL of DPPH solution (0.2 mM) were added to each test tube, fully mixed, and incubated in the dark at room temperature. The reaction took place for 30 min, and the absorbance was determined at 517 nm. Deionized water was used as the blank to replace the polysaccharide solution, and 80% anhydrous ethanol was used as the control instead of the DPPH solution. The formula was calculated as follows:$$\mathrm{DPPH}\;\mathrm{scavenging}\;\mathrm{activity}\;(\%)=(1-\;\frac{A_{sample}-A_{control}}{A_{blank}})\times 100\%$$
where *A_sample_* is the absorbance of the polysaccharide solution reacted with DPPH; *A_control_* is the absorbance of the polysaccharide solution mixed with 80% anhydrous ethanol; and *A_blank_* is the absorbance of deionized water reacted with DPPH.

#### 2.4.2. ABTS Scavenging Activity

We referred to the method of Chen et al. [18] with modifications. ABTS (7 mM) solution was mixed with aqueous potassium persulfate (2.45 mM) to generate ABTS+, and then the mixture was placed in the dark for 16 h and left on standby. The samples were prepared into 0.2, 0.4, 0.6, 0.8, 1.0, and 2.0 mg/mL polysaccharide solutions. Then, 1 mL of sample solution and 2.5 mL of freshly prepared ABTS+ solution (diluted to approximately 0.70 absorbances in PBS) were added to each test tube, and the mixture was shaken for 10 s and left for 6 min as the sample group. The blank was a group without a sample. The absorbance was measured at 734 nm, and the formula was calculated as follows:$$\mathrm{ABTS}\;\mathrm{scavenging}\;\mathrm{activity}\;(\%)=(1-\;\frac{A_{sample}}{A_{control}})\times 100\%$$
where *A_sample_* is the absorbance of sample solution reacted with ABTS+ and *A_blank_* is the absorbance of deionized water reacted with ABTS+.

#### 2.4.3. Hydroxyl Radical Scavenging

Referring to the method of Zhang et al. [19] with some modifications, the samples were prepared into 0.2, 0.4, 0.6, 0.8, 1.0, and 2.0 mg/mL polysaccharide solutions. Then, 1 mL of sample solution, 1 mL of 2 mM FeSO_4_, and 1 mL of 6 mM salicylic acid were added to each test tube and mixed well. Finally, 1 mL of 1 mM H_2_O_2_ added to the tube, 37 °C water bath for 10 min. After cooling down, the absorbance was measured at 510 nm. The formula was calculated as follows:$$\mathrm{Hydroxyl}\;\mathrm{radical}\;\mathrm{scavenging}\;\mathrm{activity}\;(\%)=(1-\;\frac{A_{sample}-A_{control}}{A_{blank}})\times 100\%$$
where *A_sample_* is the absorbance of the mixture with the sample solution; *A_control_* is the absorbance of the reaction of deionized water instead of H_2_O_2_ with the sample mixture, and *A_blank_* is the absorbance of the mixture without the sample solution.

### 2.5. Immune Activity Assay

The immune activity of rice wine polysaccharides was measured according to our previous method [14].

#### 2.5.1. Cell Culture

RAW 264.7 murine macrophage cells (purchased from Cell Resource Center, IBMS, CAMS/PUMC, Beijing, China) in logarithmic phase were collected, and the concentration of the cell suspension was adjusted to 1 × 10^5^ cell/mL. To each well of a 96-well plate, 100 μL of cell suspension was added, while the edge wells received sterile PBS instead. The cells were then incubated at 37 °C for 24 h in a cell incubator with 5% CO_2_. After the cells were plastered, the medium was replaced with serum-free DMEM and LPS (1 mg/mL), and then incubated at 37 °C in a cell incubator overnight. A 100 µL quantity of the concentration gradient (200, 100, 50, 25, and 12.5 μg/mL) of DNPs that were dissolved in the DMEM solution was added. The control group only replaced the DNP solution with DMEM, while the blank group replaced the cell suspension and the DNP solution with DMEM. There were three parallels for each group. The cells were cultured for 24 h at 37 °C in a cell incubator.

#### 2.5.2. Cell Proliferation and Cytotoxicity Assay

Cell proliferation and cytotoxic activity were detected by the CCK-8 method. The preparation of the CCK-8 working solution was as follows: CCK-8 solution was mixed with serum-free DMEM at a volume ratio of 1:10. The cells were removed from the old medium and washed twice with PBS, and a CCK-8 mixture was added to cover each sample. After incubation at 37 °C for 3 h, 100 µL of the mixture was extracted and added to each well, and a microplate reader (Labsystems Multiskan MS 352, Vantaa, Finland) was used to measure the absorbance at 450 nm.

#### 2.5.3. Determination of NO and Cytokine Content

The supernatant culture fluid was centrifuged at 1000 r/min for 10 min at 4 °C after incubation. The NO content was determined using a commercial kit (Shanghai Jianlai Biotechnology Co., Ltd., Shanghai, China) and the cytokine contents (including TNF-a, IL-1β, IL-6, and IL-10) were detected by ELISA kits (Shanghai Jianlai Biotechnology Co., Ltd., Shanghai, China). For the specific operations, refer to the instructions.

### 2.6. Statistical Analysis

The data are presented as the mean ± standard deviation. Experiments were designed with three replications. Using SPSS 21.0 software (IBM Co., Chicago, IL, USA), the significant differences (*p <* 0.05) among groups were assessed by ANOVA and one-way Duncan’s test. Graphing was performed with OriginPro 9.0 (OriginLab Co., Ltd., Northampton, MA, USA).

## 3. Results and Discussion

### 3.1. Polysaccharide Extraction

Crude polysaccharides were extracted from three groups of rice wine (DOSC, DOWA, and DOSW) and *Dendrobium officinale*, named DOSCP, DOWAP, DOSWP, and DOP, respectively. Their extraction yields were 7.26%, 8.81%, 9.14%, and 26.30%, respectively. The yield of DOP was close to that of 20.55% reported by He et al. [20]. The content of crude polysaccharides extracted from the rice wine fermented by mixed bacteria was higher than those of single bacteria (Table 2). Mixed bacterial fermentation has been suggested to combine the advantages of both bacteria and has advantages in improving the yield of polysaccharides in fermented rice wine. In addition, Table 1 shows that the sum of the polysaccharide yield from the wine broth and the fermentation residues was significantly lower than that of *Dendrobium officinale* polysaccharides (DOP). This result indicated that some *Dendrobium officinale* polysaccharides could be utilized and consumed by yeasts. Tian et al. [21] selected lactic-producing *Bacillus* DU-106 to ferment *Dendrobium officinale.* After fermentation, the polysaccharides of *Dendrobium officinale* in the stem decreased from 36.01% to 23.05%, while those in the fermentation broth increased from 0 g/L to 1.25 g/L.

### 3.2. Purification and Physicochemical Property Analysis of Polysaccharides

DOSCP, DOWAP, DOSWP, and DOP were isolated and purified by DEAE-Sepharose Fast Flow (DEAE-QFF) and Sephacryl S-300 HR columns, and the elution curves are shown in Figure 1. The fractions with the highest polysaccharide contents in each group were collected for subsequent experiments. The chemical composition of the purified polysaccharides is listed in Table 3. The total sugar content of the four polysaccharides was higher than 90%, and their protein contents were lower than 5%, indicating a higher purity of polysaccharides. All four polysaccharides comprised mannose and glucose in different proportions, and the proportion of mannose was higher in all of them. Wang et al. [22] extracted and separated DOPS-1 from the stems of *Dendrobium officinale*. DOPS-1 had a molecular weight of 1530 kDa and mainly comprised mannose, glucose, and galacturonic acid, with a mannose-to-glucose molar ratio of 2.46:1. Yu et al. [23] purified four polysaccharides from six *Dendrobium* species with Mw values of 1341.06 kDa, 540.22 kDa, 1415.64 kDa, and 1320.26 kDa. Similar to a previous study, the DOP polysaccharide used in this study was extracted from *Dendrobium officinale* using hot water extraction, obtaining a Mw of 1268.21 kDa and containing mannose and glucose in a ratio of 3.29:1. In DOWAP, the mannose-to-glucose ratio was highest, while it was lowest in DOSWP. No significant differences were found between DOSCP and DOP. The fermented polysaccharides DOSCP, DOWAP, and DOSWP were 25.74 kDa, 15.01 kDa, and 17.67 kDa, respectively (Appendix A). According to the findings, the Mw of polysaccharides significantly decreased after fermentation. However, only their relative proportion and not their composition were altered.

Fermentation can change the monosaccharide composition, Mw, and structure of polysaccharides, and the specific changes may be related to fermentation conditions, strains of bacteria, and so on. Li et al. [11] reported that fermentation of *Dendrobium officinale* polysaccharides by lactic acid bacteria resulted in noticeable alterations in the polysaccharide molecular weight distribution, amount of total sugars and reducing sugars, and the molar ratio of monosaccharides. Tian et al. [21] used lactic acid-producing *Bacillus* DU-106 to ferment *Dendrobium officinale*, and the Mw of *Dendrobium officinale* polysaccharides was 4.92 × 10^5^ Da before fermentation, 5.21 × 10^5^ Da after fermentation, and 4.15 × 10^5^ Da in the fermentation broth.

### 3.3. Antioxidant Activity

The scavenging abilities of DOSCP, DOWAP, DOSWP, and DOP against DPPH radicals, ABTS radicals, and hydroxyl radicals were investigated separately to evaluate the antioxidant activity of the four polysaccharides. Vc was used as a positive control. These data are illustrated in Figure 2A. The scavenging ability of DPPH radicals showed a significant positive correlation within the range of 0.2–2.0 mg/mL. At a concentration of 2.0 mg/mL, the scavenging activities of the four polysaccharides were 26.55% (DOSCP), 32.24% (DOWAP), 33.70% (DOSWP), and 29.91% (DOP). The polysaccharide of rice wine fermented by mixed bacteria (DOSWP) had the strongest scavenging ability against DPPH radicals, but that of DOSCP was the weakest, followed by that of *Dendrobium officinale* polysaccharides (DOP). 

The polysaccharides in each group had strong scavenging ability against ABTS radicals (Figure 2B). At the same concentration, the scavenging ability of DOWAP was significantly higher than that of the remaining three groups of polysaccharides, and the scavenging ability against ABTS radicals of unfermented polysaccharide DOP was the weakest. The scavenging activity of each group of polysaccharides against ABTS radicals increased significantly in the range of 0.2~1.0 mg/mL, and the growth trend of the scavenging activity slowed after the polysaccharide concentration exceeded 1.0 mg/mL. When the polysaccharides reached 2.0 mg/mL, the maximum scavenging activity of the four polysaccharides was 38.26% (DOSCP), 54.06% (DOWAP), 38.26% (DOSWP), and 21.30% (DOP). 

Figure 2C reveals the effect of each polysaccharide on scavenging hydroxyl radicals. Within the range of 0.2–2.0 mg/mL, polysaccharides also demonstrated an increasing trend in scavenging hydroxyl radicals. The scavenging rates of hydroxyl radicals by DOP and DOSP were similar in the range of 0.2~2.0 mg/mL, and DOP was more effective. The scavenging rates of the four polysaccharides at a polysaccharide concentration of 2.0 mg/mL were 89.88% (DOSCP), 86.79% (DOWAP), 95.66% (DOSWP), and 91.53% (DOP).

In summary, DOSWP showed a relatively high scavenging activity of DPPH and hydroxyl radicals, and DOWAP showed the best scavenging effect on ABTS radicals, which was generally better than the *Dendrobium officinale* polysaccharides before fermentation. The antioxidant activity of polysaccharides can be significantly improved by fermentation, and it is closely related to their structural characteristics and may depend on factors, such as monosaccharide composition, molecular weight, glycosidic bonds, chain conformation, and degree of branching [24]. The polysaccharides before and after fermentation tend to exhibit different biological activities, and the polysaccharides are more active after fermentation. Ai et al. [9] applied *Saccharomyces cerevisiae* to ferment ginseng, and the antioxidant activity of fermented ginseng polysaccharides was higher than that of unfermented ginseng polysaccharides. Chen et al. [10] used *Bacillus subtilis* and *Saccharomyces cerevisiae* to ferment two wheat bran polysaccharides, and the scavenging ability of DPPH radicals and hydroxyl radicals of the polysaccharides was increased after fermentation. Previous studies have reported that at the same mass concentration, polysaccharides with lower relative molecular mass usually have more free hydroxyl groups and higher reducing sugar content, thus exerting a stronger antioxidant capacity [25]. The monosaccharide compositions of the four polysaccharides in this study were similar. However, the molecular weights of polysaccharides after fermentation in rice wine were significantly lower than those of unfermented *Dendrobium officinale* polysaccharides, which may be one of the reasons why the antioxidant activity of the fermented polysaccharides in this study was mostly higher than those of unfermented polysaccharides. Active polysaccharides with enhanced antioxidant activity can be obtained by fermentation of *Saccharomyces cerevisiae* and *Wickerhamomyces anomalous*.

### 3.4. Anti-Inflammatory Activity

#### 3.4.1. Cell proliferation and Cytotoxicity Assessment

Macrophages play an important role in host defense and the innate immune response [26]. However, before studying their anti-inflammatory activity in vitro, it is necessary to evaluate the effect of DNP on the proliferative activity and cytotoxicity of RAW264.7 cells. The results, as shown in Figure 3A, showed that all four polysaccharides DOP, DOSWP, DOWAP and DOSWP had cell proliferative effects on RAW264.7 cells in a concentration-dependent manner. The proliferative effect was enhanced with increasing polysaccharide concentration. The survival rates of cells treated with DOP, DOSWP, DOWAP, and DOSWP were 287.89 ± 9.41%, 251.05 ± 9.57%, 265.53 ± 12.49%, and 261.32 ± 3.62%, respectively, when the polysaccharide concentration reached 200 μg/mL. There was a significant proliferative effect (*p* < 0.05). DNP1 and DNP2 had significant proliferative effects on RAW264.7 cells in the range of polysaccharide concentrations of 12.5–200 μg/mL (*p* < 0.05), indicating that none of the four polysaccharides was cytotoxic.

#### 3.4.2. Effect of Polysaccharides on the Content of NO and Cytokines

Particular immunological processes, especially the immune function of macrophages, can be activated in response to invasion by foreign substances. The RAW 264.7 cell line in mice is a mononuclear macrophage that has strong antigen adsorption and phagocytosis ability when cultured in vitro. It can release relevant immunomodulatory mediators in response to antigen stimulation, which stimulates the immunomodulatory role of macrophages in vivo and to a certain extent. RAW 264.7 cells are frequently employed as a model for research on in vitro immune function [27]. Polysaccharides can induce macrophages to generate NO, TNF, and interleukin factors to improve the immune response of the host [28].

Lipopolysaccharide (LPS), an endotoxin derived from the outer membrane of gram-negative bacteria [29], is a natural immune and inflammatory stimulator. It has been reported to trigger inflammatory responses by activating a series of intercellular signaling pathways that lead to the release of proinflammatory cytokines by cells [30]. Therefore, in this study, inflammation experiments were performed using LPS-induced RAW 264.7 cells and NO and cytokine levels were assessed using NO and ELISA kits.

NO is a crucial inflammatory compound that plays a role in the control of the immune response [31]. We examined the effect of four polysaccharides on macrophage NO production and explored the anti-inflammatory effects of the four polysaccharides. As shown in Figure 3B, the NO content of RAW 264.7 cells was significantly increased after incubation with LPS (1 μg/mL), reaching 47.71 ± 1.56 μg/mL. NO production by RAW 264.7 cells was significantly reduced after treatment with different doses of polysaccharides in a concentration-dependent manner. When the concentrations of DOP, DOSWP, DOWAP, and DOSWP were 200 μg/mL, the NO contents were 6.70 ± 0.77 μg/mL, 7.53 ± 0.61 μg/mL, 9.30 ± 2.70 μg/mL, and 7.38 ± 0.51 μg/mL, respectively, indicating that the four polysaccharides significantly inhibited the NO production induced by LPS stimulation.

TNF-α is a cytokine that promotes inflammation and is responsible for various intracellular signaling events, including cell necrosis and apoptosis. It also stimulates the release of cytokines including IL-6 and IL-1β [30]. In inflammation models, decreasing the levels of IL-1β, IL-6, and TNF-α can reduce the inflammatory response and suppress the immune response [23]. As shown in Figure 3C, the TNF-α content increased 15.34-fold to 77.61 ± 5.95 pg/mL under LPS stimulation. Compared with the LPS group, the treatment with the four polysaccharides inhibited the LPS-stimulated TNF-α production by RAW 264.7 cells. When the concentration of the four polysaccharides was 12.5–200 μg/mL, the TNF-α production by RAW 264.7 cells was significantly lower than that of the LPS group, in which there was no significant difference in the inhibitory effect of the four polysaccharides on TNF-α production by RAW 264.7 cells.

The inflammatory mediator IL-1β plays a role in the initiation and escalation of the inflammatory response leading to intestinal injury [32]. Figure 3D shows that IL-1β levels in the LPS group reached 27.13 ± 0.20 pg/mL, which was 38.06 times higher than that in the control group, while the IL-1β levels in the polysaccharide group were significantly inhibited. When the doses of DOP, DOSWP, DOWAP, and DOSWP were 200 μg/mL, the IL-1β levels in the DOP, DOSWP, DOWAP, and DOSWP groups were 3.06 ± 0.39 pg/mL, 4.04 ± 0.33 pg/mL, 5.88 ± 0.41 pg/mL, and 3.19 ± 0.12 pg/mL, respectively, which were significantly lower than those in the LPS group. In addition, elevated IL-6 is considered a typical sign of an acute inflammatory response. lPS stimulated a significant increase in IL-6 production compared to the blank group. In contrast, the addition of four *Dendrobium* polysaccharides significantly inhibited IL-6 production in RAW 264.7 cells (*p* < 0.05) in a concentration dependent manner (Figure 3E).

IL-10 is an anti-inflammatory factor that effectively regulates the cellular inflammatory response [26]. The effect of polysaccharides on IL-10 secretion by LPS-treated RAW 264.7 macrophages is shown in Figure 3F. When the inflammatory cells were treated with the four polysaccharides separately, the content of IL-10 was significantly increased, and the release of IL-10 increased with increasing polysaccharide concentration. Moreover, the effect of fermented polysaccharides (DOSCP, DOWAP and DOSWP) on promoting IL-10 release from LPS- induced macrophages RAW 264.7 cells was more significant than that of unfermented polysaccharides (DOP) at low polysaccharide concentrations (12.5–25 µg/mL). However, the trend was reversed at high polysaccharide concentrations (100–200 µg/mL). 

In the present study, all four polysaccharides showed good inhibitory effects on the LPS-induced release of inflammatory factors from RAW 264.7 macrophages. However, there was no significant variability in the anti-inflammatory activity of the four polysaccharides. The molecular weight of the polysaccharides changed significantly before and after fermentation, but the effect on the anti-inflammatory activity of the polysaccharides was minimal. This may be because the direct regulation of inflammation by polysaccharides relies mainly on their action on TLRs on the cell surface rather than their penetration into host cells [33].

## 4. Conclusions

In the present study, we investigated the effects of mono-fermentation (*Sc* and *Wa*) and co-fermentation (*Sc+Wa*) on the physicochemical properties and bioactivity of *Dendrobium officinale* polysaccharides. The polysaccharides in rice wine before and after the fermentation of *Dendrobium officinale* were extracted and purified to homogeneous fraction polysaccharides and their chemical fraction, monosaccharide composition, Mw, in vitro antioxidant capacity, and immunomodulatory capacity were analyzed. The results showed that the extraction yields of crude polysaccharides in rice wine were 7.26%, 8.81%, and 9.41% for DOSCP, DOWAP, and DOSWP, respectively. The yield of *Dendrobium officinale* crude polysaccharides was 26.30%. Furthermore, the molecular weights of the purified fractions DOSCP, DOWAP, DOSWP, and DOP were 25.73 kDa, 15.01 kDa, 17.67 kDa, and 1268.21 kDa, respectively. The extraction yield and molecular weight of the polysaccharides were significantly reduced after fermentation. Moreover, all four polysaccharides comprised mannose and glucose, with ratios of 3.31:1, 5.56:1, 2.40:1, and 3.29:1. The four polysaccharides had good scavenging activities against DPPH radicals, ABTS radicals and hydroxyl radicals, and the scavenging effect of the fermented polysaccharides on free radicals was better than that of the unfermented polysaccharides. Additionally, all four polysaccharides were able to inhibit the release of NO, TNF-α, IL-6, and IL-1β and promote the release of IL-10, thereby reducing the inflammatory response of LPS-induced macrophages RAW 264.7 macrophages in mice. However, there were no significant differences in the impact of the four polysaccharides on cell proliferation activity. The results indicated that fermentation could significantly reduce the molecular weight of *Dendrobium officinale* polysaccharides and enhance the antioxidant activity of *Dendrobium officinale* but had little effect on the anti-inflammatory activity. This study provides a new reference method for antioxidant enhancement of polysaccharides.

## Figures and Tables

**Figure 1 foods-12-00150-f001:**
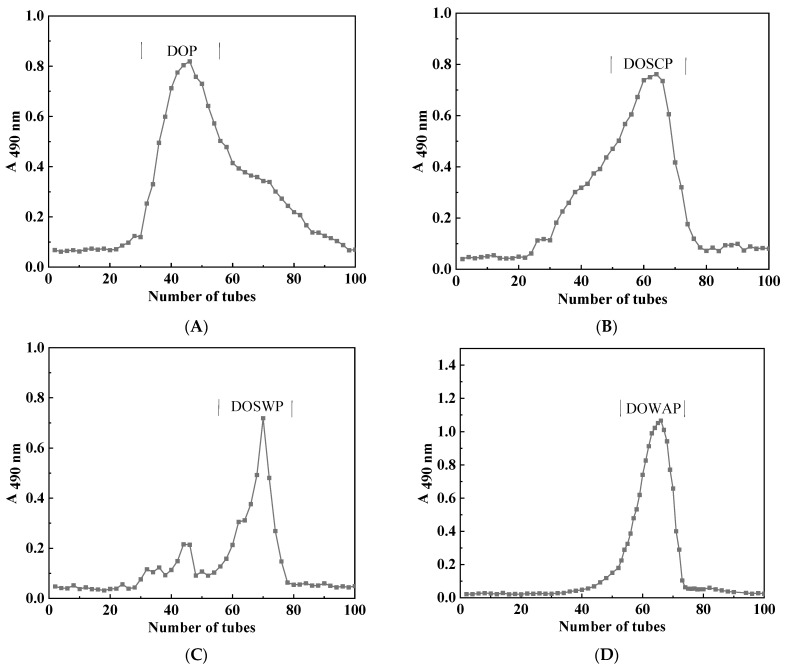
Elution curves of polysaccharides. (**A**) DOSCP, (**B**) DOWAP, (**C**) DOSWP, and (**D**) DOP.

**Figure 2 foods-12-00150-f002:**
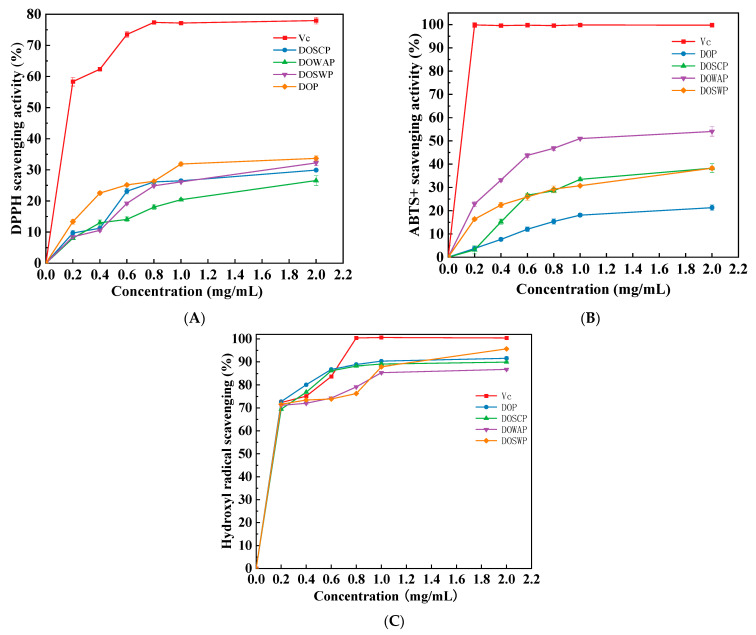
DPPH radical scavenging activity of polysaccharides (**A**); ABTS free radical scavenging activity of polysaccharides (**B**); hydroxyl radical scavenging activity of polysaccharides (**C**).

**Figure 3 foods-12-00150-f003:**
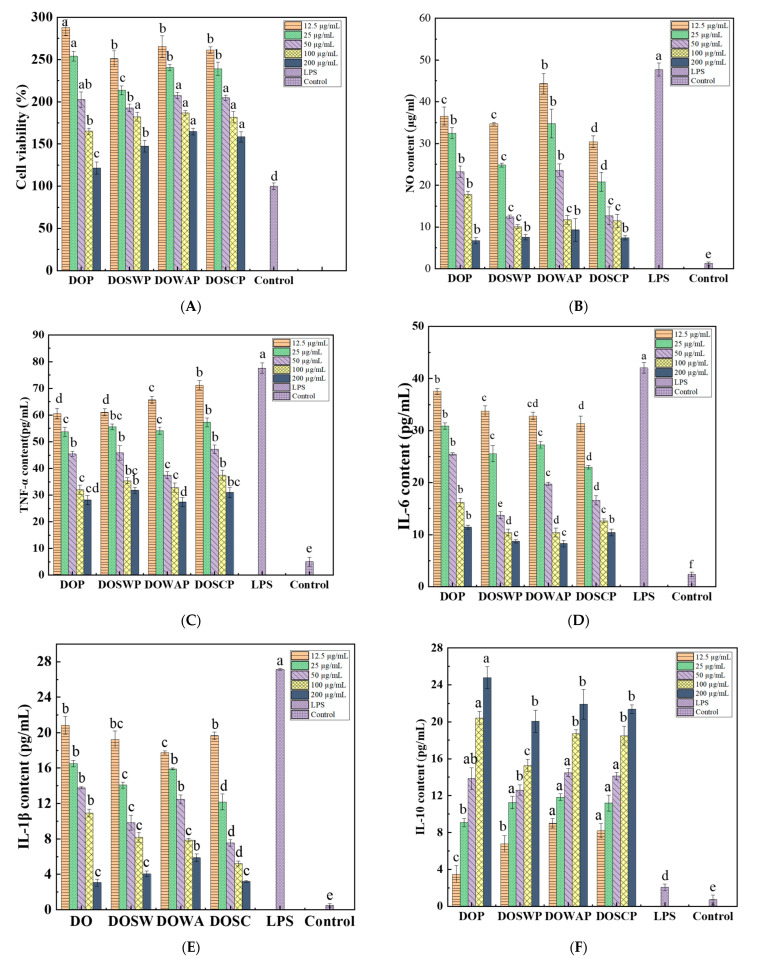
Effect of polysaccharides on the proliferation of murine macrophage RAW 264.7 cells (**A**); Effect of polysaccharides on the content of NO (**B**), TNF-α (**C**), IL-6 (**D**), IL-1β (**E**), IL-10 (**F**) in murine macrophage RAW 264.7 cells. Different lowercase letters above the columns indicate significant differences among groups (*p* < 0.05).

**Table 1 foods-12-00150-t001:** Polysaccharides extraction method.

Samples	Extraction Method	Yeast
DOSCP	fermentation	*Sc*
DOWAP	fermentation	*Wa*
DOSWP	fermentation	*Sc+Wa*
DOP	Hot water	-

**Table 2 foods-12-00150-t002:** The yields of polysaccharides.

Samples	Yield (%)	Samples	Yield (%)
DOSCP	7.26 ± 0.23 ^c^	DOSCPR	0.78 ± 0.28 ^e^
DOWAP	8.81 ± 0.23 ^bc^	DOWAPR	0.72 ± 0.06 ^e^
DOSWP	9.41 ± 0.10 ^b^	DOSWPR	1.79 ± 0.65 ^d^
DOP	26.30 ± 1.83 ^a^		-

Different lowercase letters indicate significant differences among groups (*p* < 0.05).

**Table 3 foods-12-00150-t003:** Mw, monosaccharide composition and chemical composition.

Samples	M_W_/kDa	Mannose/%	Glucose/%	Mannose-to-Glucose	Total Sugar (%)	Protein (%)
DOSCP	25.74 ± 0.21 ^b^	75.71 ± 1.95 ^b^	24.29 ± 1.95 ^b^	3.31	92.86 ± 1.97 ^a^	4.25 ± 2.02 ^a^
DOWAP	15.01 ± 0.53 ^c^	84.75 ± 0.12 ^a^	15.25 ± 0.12 ^c^	5.56	93.73 ± 1.96 ^a^	1.60 ± 0.30 ^b^
DOSWP	17.67 ± 0.34 ^bc^	70.55 ± 0.63 ^c^	29.45 ± 0.63 ^a^	2.40	90.30 ± 0.66 ^a^	4.54 ± 0.19 ^a^
DOP	1268.21 ± 10.23 ^a^	76.67 ± 0.60 ^b^	23.33 ± 0.60 ^b^	3.29	92.02 ± 2.52 ^a^	1.06 ± 1.15 ^b^

Different lowercase letters indicate significant differences among groups (*p* < 0.05).

## Data Availability

Data is contained within the article.

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
