# Peer review of "Effect of Yeast Fermentation on the Physicochemical Properties and Bioactivities of Polysaccharides of Dendrobium officinale"

_foods, 2022, doi:10.3390/foods12010150_

Round 1

Reviewer 1 Report

Dear authors and editors’ thanks for your work and give me chance to read and revise the manuscript entitled “Effect of yeast fermentation on physicochemical properties and bioactivities of polysaccharides of Dendrobium officinale. I have some comments that should be done to improve your manuscript.

First of all the writing style needs to be improved more and more.

Abstract:

I would the authors to modify and follow this structure in the abstract; background and problem, the rationale for the study; research objectives; some methodology; important data including statistical analysis; conclusions, novelty, and the importance of the findings. Also, please add the full name of the yeasts.

Acronyms/Abbreviations:

should be defined the first time (and only one time) they appear in each of three sections: the abstract; the main text; the first figure or table (when defined for the first time, the acronym/abbreviation should be added in parentheses); then use acronyms.

Introduction:

Line 76: Please add one paragraph explaining how the applied yeasts work in the food. What are their benefits?

Line 84: rewrite the aim of your work clearly with more details and how it is good for a food science major regarding food microbiology and fermentation process.

Materials and Methods:

In the chemicals section, please mention all the chemicals and solvents you used in more detail and follow the journal format to write the chemical information.

Materials:

For Dendrobium officinale, I would the authors to add more information about it. Not only the place they obtained from it.

Do the authors analyse the raw materials before they are used in the experiments?

Preparation and fermentation conditions

For the fermentation process for the conditions, why did the authors select these parameters? Did they do pre-experiments? Or there is a reference? What is the reason for these selected parameters?

What about the safety of Saccharomyces cerevisiae FBKL2.8022 and Wickerhamomyces anomalous FBKL2.8023 ? Did the authors do it? If they did it in a previous study, please mention the reference.

For the experiment design, I suggest the authors make a table mentioning the groups with the condition of each one.

Immune activity assay

Explain the method in brief. I know you mentioned the reference.

Results and discussion:

I would the authors to try to make the discussion in deep with the previous studies.

Throughout the whole manuscript, the authors should write the scientific names of the microorganisms or the plants in italic form. Please check and revise.

Author Response

Response to Reviewer 1 Comments

Dear reviewer:

Thank you for your comments and suggestions. We also appreciate your efforts in reviewing our manuscript. All these comments and suggestions are valuable and helpful to us in revising and improving the manuscript. This manuscript " Effect of yeast fermentation on the physicochemical properties and bioactivities of polysaccharides of Dendrobium officinale" has been revised based on your comments and suggestions. The point-by-point replies are as follows:

Point 1: Abstract: I would the authors to modify and follow this structure in the abstract; background and problem, the rationale for the study; research objectives; some methodology; important data including statistical analysis; conclusions, novelty, and the importance of the findings. Also, please add the full name of the yeasts.

Response 1: We have modified as suggested.

Point 2: Acronyms/Abbreviations: should be defined the first time (and only one time) they appear in each of three sections: the abstract; the main text; the first figure or table (when defined for the first time, the acronym/abbreviation should be added in parentheses); then use acronyms.

Response 2: We have modified as required.

Point 3: Introduction:

Line 76: Please add one paragraph explaining how the applied yeasts work in the food. What are their benefits?

Line 84: rewrite the aim of your work clearly with more details and how it is good for a food science major regarding food microbiology and fermentation process.

Response 3: We have added the relevant introduction in the article (lines 82-98).

Point 4: Materials and Methods: In the chemicals section, please mention all the chemicals and solvents you used in more detail and follow the journal format to write the chemical information.

Response 4: We have revised it as requested.

Point 5: Materials: For Dendrobium officinale, I would the authors to add more information about it. Not only the place they obtained from it.

Do the authors analyse the raw materials before they are used in the experiments?

Response 5: The Dendrobium officinale treatment has been added as requested (line 102). No previous analysis has been done, mainly referring to the results of previous studies. It was used as the control group in this study.

Point 6: Preparation and fermentation conditions

For the fermentation process for the conditions, why did the authors select these parameters? Did they do pre-experiments? Or there is a reference? What is the reason for these selected parameters?

What about the safety of Saccharomyces cerevisiae FBKL2.8022 and Wickerhamomyces anomalous FBKL2.8023 ? Did the authors do it? If they did it in a previous study, please mention the reference.

For the experiment design, I suggest the authors make a table mentioning the groups with the condition of each one.

Response 6: (1) Pre-experiments were done and also a relevant part of the experimental conditions screened for the process optimization of rice wine. It is the optimization of active substance content, physicochemical index and sensory quality, which is not very relevant to the content of this article, so it was not put into the article. (2) The brewer's yeast was screened from xiaoqu by Prof. Wang Chunxiao, and the two yeast strains have a low yield of higher alcohols and a high or medium ethanol production capacity, which is safer compared to other brewer's yeasts, corresponding to the reference [13]. (3) The experimental design has been added to Table 1.

Point 7: Immune activity assay

Explain the method in brief. I know you mentioned the reference.

Response 7: We have added experimental methods and materials (section 2.5).

Point 8: Results and discussion:

I would the authors to try to make the discussion in deep with the previous studies.

Response 8: We've revised this section to in-depth analysis and discussion.

Point 9: Throughout the whole manuscript, the authors should write the scientific names of the microorganisms or the plants in italic form. Please check and revise.

Response 9: We have checked and modified as required.

Reviewer 2 Report

THis work is not novel.

Errors bars are missing in Fig. 2

Discussion is still poor and needs extensive revision

Authors have not written conclusions well 

Reviewer 3 Report

The manuscript"Effect of yeast fermentation on physicochemical properties and bioactivities of polysaccharides of Dendrobium officinale" has been reviewed and found that, the manuscript needs and extinsive improvment and the following comments must be consided from the authors.

1. The  English language, the senctences structure and grammer need  intensive revision and  improvement 

2. Lines 104&105, the source of enzymes must be added

3. lines 107, 108, why the temperature (30C ) has been used for formentation. why the authors didn't conducted the fermentation on various tempraturs and other frementation conditions to optain the optimum conditions?

4. in section 2.3. the extraction of polysaccharides, why the authors extracted polysaccharides from seeds powder broth media with the same method? the extraction methods from powder must be differes from liquids 

5. lines 117-120, the centrifigation speed (r/min) must be changed to (rpm or xg)

6. The model, manufacturer, city and country of manufacturer of all used apparatuses must be presented in materials and methods

7.  in section 2.4, which typr of column has been used in the experiment (which matrix) fo dtermination of MW and which used for determination the simple sugars composition? this section of methodology must be impovedand supported with details to be more clearer 

8. 2.6 "Immune activity assay", the details of methods must be provided.

9. lines 215-216 "All four polysaccharides were composed of mannose and glucose with different proportions, and the proportion of mannose was higher in all of them" from which analysis the authors got these results? explain the presedure to get the monosacchrides content.

10. Fig1 did not appear the MW of different polysaccharides? from wher the authors got the MWs? please explain in the manuscript.

11. In fig2, vc must be identified in the figure key, the "content" must be changed to " concentration"

12. section 3.4.  Anti-inflammatory activity was discussed without any experimental data. the data and results obtainedfrom the experiment must be presented and interpreted. 

13. Anti inflamatory activity was presented in results and discussion but didn't present in materials and methods.

14. section 3.4.1. Cell proliferation and cytotoxicity assessment, the results poor discussed and interpreted. the results should be interpreted based on the obtained results 

15. 3.4.1. Cell proliferation and cytotoxicity assessment did not presented in the materials and methods section.

16. Cell proliferation and cytotoxicity assessment should be presented in materials and methods section with an appropriate experimental design.

17.  section of 3.4.2. the interpretation of the Effect of polysaccharides on the content of NO and cytokines is unclear and needs an extensive revision

18. The Effect of polysaccharides on the content of NO and cytokines must be presented in materials and methods

19. the conclusion needs and extensive improvement with focusing on the novelty that supported with and results. furthermore, the conclusion must show the importance of the findings for the field.

20. overall, the manuscript needs a deep revision and the interpretationof the results must be improved. 

Round 2

Reviewer 1 Report

Accept

Author Response

Thank you for your comment.